# Implementation of SDGs in University Teaching: A Course for Professional Development of Teachers in Education for Sustainability for a Transformative Action

**Leslie Mahe Collazo Expósito \* and Jesús Granados Sánchez** 

Department of Subject-Specific Education, University of Girona, 17004 Girona, Spain; Jesus.granados@udg.edu
\* Correspondence: leslie.collazo@udg.edu; Tel.: +34-638-705-827

**Abstract:** The University Jaume I of Castellon (Spain) launched the "*ImpSDGup*" course in 2017. The aim of this training course on professional education for sustainable development (ESD) skills and competences for higher education teachers is to help academics in reorienting their subjects' curricula in order to contribute to the sustainable development goals of the United Nations' Agenda 2030. The "*ImpSDGup*" course is based on the Training Model in Transformative Action for Sustainability (TMTAS) model that is structured in three main areas: the content on sustainability, the theoretical approaches, and the ESD teaching and learning methodologies. In this paper, we describe the design and the contents of the training course and we investigate the changes that participants of the course implemented in the design of their subject programs. The methodology of the study was based on content analysis. The main results show that most of the 55 university teachers that undertook the course in its three editions succeeded in transforming course objectives and that they incorporated some of the SDGs and several ESD active learning methodologies in their teaching. As a consequence of this, we consider that the course helps in enhancing university teachers' ESD professional competences.

**Keywords:** teacher training; sustainable development goals; education for sustainability; curriculum reorientation; higher education

## 1. Introduction

The events that mark and condition our lives today make us perceive the expression "planetary emergency" [1] as something close in time, and therefore the application of sustainability principles is of paramount importance. Authors such as Schumacher [2] and Tilbury [3] point out that ecological, economic, and social problems continue to grow, despite the increase of knowledge on sustainability. In order to solve major unsustainable problems, the United Nations launched the Agenda 2030 with its 17 sustainable development goals (SDGs) [4], with higher education institutions needing to be at the front line in terms of contributing to the generation of sustainable practices, improving the ways sustainability is perceived, taught, modeled, and implemented. Different reports [5,6] present various analysis of how higher education institutions have incorporated educational practices based on sustainability, finding that major progress has been made in actions related to the environmental management of their campuses, or the creation of green structures. However, a sustainable university must be committed to sustainability in more than campus greening programs [7] and must include proper education and training, involve new ways of doing research, and promote an authentic engagement with the community. Support for universities in the transition to a sustainable paradigm is being promoted by organizations such as the Sustainable Development Solutions Network (SDSN),

a global initiative of the United Nations that mobilizes the experience and resources of academia, civil society, and the private sector, providing solutions for sustainable development at local, national, and global levels. One of these solutions is the guide "How to get started with the Sustainable Development Goals (SDGs) in universities" [8], which facilitates the transition of universities towards sustainability starting with the SDGs. This work is complemented with the Global University Network for Innovation (GUNi) reflection on the challenges, obstacles, and opportunities that universities face in the implementation of the SDGs [9].

Education for sustainable development (ESD) must play a central role in our unavoidable commitment to build a sustainable future for the good of our society and the planet [10]. One of the key areas of ESD in higher education is the reorientation of the curriculum towards sustainability. The Sectoral Commission for Environmental Quality, Sustainable Development and Risk Prevention (CADEP) of the Conference of Rectors of Spanish Universities (CRUE) has named this process as "curricular sustainability" [11], while others prefer to refer to it as sustainable curriculum [7] or curriculum reorientation [12]. Although it is not a new topic in Spain nor internationally [13], it is well known that it is a concept that has not been applied in all universities and disciplines [14] and that needs further development. To advance in this challenge, higher education institutions must implement changes in areas such as leadership; management; research; and, mainly, in the training of teachers [5]. In this sense, many international projects such as University Teachers for Sustainable Development (*UE4SD*) [15] have been developed in recent years for the improvement of teachers' competences in ESD. At a smaller scale, we can find a diverse range of university initiatives, starting from the creation of frameworks and training models to embed ESD [16,17] and guidelines on how to address the SDGs [18]. In this area, we can locate our work presented here—the course *Implementation of Sustainable Development Goals for University Teachers (ImpSDGup)*, a teacher training course that has been designed to guide the integration of ESD into the curriculum of university subjects. *ImpSDGup* is based on the *Training Model in Transformative Action for Sustainability* (TMTAS), which provides an overall theoretical framework for designing transformative ESD actions. The *ImpSDGup* course has been delivered three times at the University Jaume I (UJI) from 2017 to 2019, with a participation of 55 university teachers thus far.

This paper has two main aims:

- The first aim is to describe and theoretically justify the design and delivery of the *ImpSDGup* course as a teacher training model to guide the integration of ESD into the curriculum of university subjects.
- The second aim is to investigate what ESD transformations participants of the *ImpSDGup* course implement in the design of the curricula and programs of their course subjects.

According to these two aims, the paper is structured in two different but complementary parts: the first part includes a theoretical framework on sustainability in higher education curricula and the design of the *ImpSDGup* training course. The second part of the paper presents the research design and the findings of the study.

## 2. Theoretical Framework: The Reorientation of Curricula towards Sustainability in Higher Education

In this section we explore the following two questions: How can we reorient our university courses and embed ESD? How can university teachers be trained in this field?

ESD provides us with concepts, principles, values, skills, and competences that are highly important for the co-creation of our sustainable collective future. ESD started as such after Agenda 21 [19]—its chapter 36 was a recognition of the role of education in achieving sustainability and entitled all the countries of the world to reorient their national education systems towards ESD. Considering time, the degree of change, and the level of institutional commitment, Granados-Sanchez et al. [20] suggest that the higher education institutions (HEIs) pathways towards sustainability could be

characterized by continuity, radical change, or transition. Transition could be also split in two different scenarios: one where continuity and transition coexist, and another where partial transition leads to a complete transformation of the institution. For Filho [21], there are three main approaches being used by higher education institutions when implementing sustainability and ESD: the individual approach (tackled by individual academics); the sectorial approach (a department or faculty); and the institutional approach, where the whole university is committed to sustainability. The implementation, then, could be a top-down or a bottom-up process [22]. The integration of sustainability could also maintain or change current degree structures and focus on working through disciplines and subjects or through an interdisciplinary approach [23]. Moreover, even a *glocal* curriculum could be negotiated and implemented through transnational collaboration [24].

Since the mid-1990s, universities initiated an embedding of sustainability and ESD in their curricula, and we can acknowledge a considerable progress in the field. However, there are still some areas that need attention and improvement. For example [21], universities must create institutional guidelines for sustainability; they must promote academic engagement and training in sustainability that leads to curriculum reorientation and innovation; they also must work in reducing their carbon footprint; and they must develop approaches for the integration of teaching, research, management, and community engagement. ESD should not be considered as an add-on to the current curriculum of subjects by adding a disconnected sustainability theme. It implies something more—for Sterling [17], it is essential for curriculum reorientation that subjects are designed with a focus on ESD by introducing its concepts, content, values, competences, and teaching methodologies. We think this could be a good starting point, but it is not enough if we want to be fully sustainable—the university is a complex system where different institutional areas (such as research, management, and community engagement) interact and where ESD apply to all of them.

Taking into account different proposals [25–29], we consider that the main criteria for embedding sustainability in curricula are related to the following competences:

- To understand how professional activity contributes or affects the sustainable development of the environment, society, and the economy, at all scales, in order to identify challenges and prevent risks and impacts.

  This implies learning to act ethically, to clarify one's own values, and to ask critical questions.

- To actively participate and work with others in interdisciplinary and transdisciplinary teams to achieve sustainable development.

  This competence enables people to participate in discussions and intercultural dialogues, as well as to collaborate with other stakeholders in the design and implementation of actions and policies that can transform reality. In action-oriented learning, learners participate in actions and reflect on their experiences in terms of the desired learning process and their personal development. The experience can come from a project (service learning), an internship, the holding of a workshop, the implementation of a campaign, and so on. Action learning is based on Kolb's theory of the empirical learning cycle with the following stages [30]: (a) to have a concrete experience, (b) to observe and reflect, (c) to understand abstract concepts and make generalizations, and (d) to apply them to new situations.

- To develop and apply complex and systems thinking.

  Bonill et al. [31] exemplify how training models committed to the perspective of complexity should reflect a systemic, dialogical, and hologrammatic approach. The incorporation of the systemic perspective implies situating the phenomena under study as organizations, where a multitude of causes and effects converge simultaneously, with a significant component of indetermination, and considering the temporal dimension that gives relevance to the evolutionary and historical perspective of the phenomena. The dialogical perspective consists in rethinking the approach to phenomena by means of constant dialogue of opposites (between order and disorder, stability and change, and so on). Finally,

the hologrammatic vision implies the constant scalar analysis (the macro, meso, and micro) without losing sight of the interconnections among them.

- To solve current problems and to envision and create sustainable futures by transformative actions.

ESD should provide transformative learning, which can be defined better by its objectives and principles rather than by a teaching or learning strategy. In this approach, the teacher is a facilitator who empowers learners to challenge and change unsustainable practices [32]. This approach is complemented by the related concept of transgressive learning [33] that goes one step further—it emphasizes that learning in ESD has to overturn the status quo and prepare the learner for disruptive thinking and co-creation of new knowledge. Envisioning change implies (a) learning from the past—it includes critical analysis and thorough understanding of past developments, including the root causes of those developments. It draws lessons through understanding both successes and failures in cultural, social, economic, and environmental spheres; (b) inspiring active engagement in the present; (c) exploring alternative futures—ESD should explore and lead to positive futures for people and nature [34]. This process draws upon scientific evidence, uncovers current beliefs and assumptions that underlie our choices, and encourages creative thinking. This offers ownership, creativity, direction, and energy that can motivate people to make transformative actions for sustainability.

*The TMTAS Framework*

The TMTAS is a theoretical framework developed within the PhD thesis of the lead author [35] of this paper, which proposes that ESD university teacher training should focus on three main aspects: the contents on sustainability, the theoretical approaches, and the ESD teaching methodologies (see Figure 1).

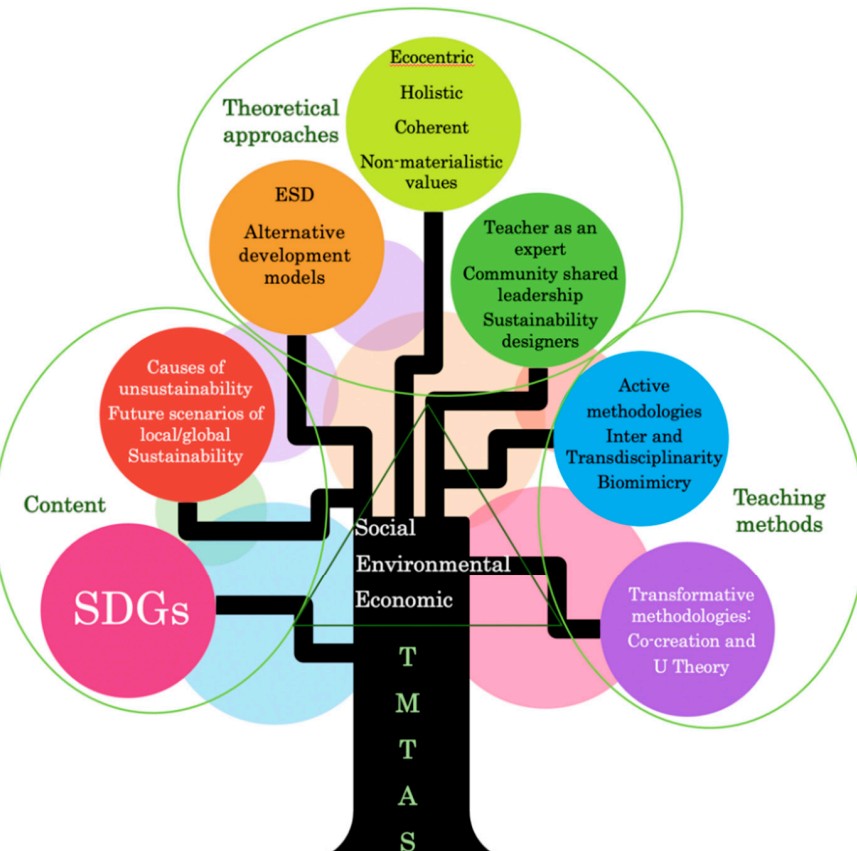

**Figure 1.** Training Model in Transformative Action for Sustainability (TMTAS).

Regarding the contents of sustainability, Agenda 21 [19] set out the main list of issues and themes to date, but today Agenda 2030 and the SDGs [4] must be at the forefront of our actions. All the content that emerge from the SDGs should include the three dimensions of sustainability (the environmental, the social, and the economic) and the relationships among them. The contents must have a retrospective and a prospective approach, and this implies an investigation of the root causes of unsustainability, as well as an imagination of future scenarios through the knowledge of new sustainable proposals and good practices that are developed around the world and that allow us to go beyond current trends and boundaries.

The theoretical approaches respond to the recognition of the need to move from the current anthropocentric worldviews and patterns of development to a biocentric and/or ecocentric worldviews. This transformation highlights the role of emotions and intuition in applied science and places environmental ethics (and values such as compassion, imagination, creativity, and spirituality) in a privileged position. It is time for the connection between matter and consciousness, the link between quantum physics and its ontological implications, and to see nature as an "extended mind". To this end, the challenges of building a sustainable education system include the coherence between discourse and practice, the integration of a holistic approach, and the attribution to teachers of the role of trainer of sustainability designers [36].

Teaching methodologies and strategies are a very important component of subject design. They must be coherent with the acquisition of sustainability competencies by learners and the way they are assessed. For us, the main teaching methodologies for sustainability should include the following: scientific learning (to practice with the scientific method); case study, problem-based learning, project-based learning, and service learning (to experiment with interdisciplinarity and transdisciplinarity, and to gain insights from different perspectives); contextualized learning and experiential learning (so students learn from their own experience and connect what they already know with new information); artistic expression techniques, role-playing, and storytelling (that allow students to express their thoughts and emotions visually and symbolically); simulations; cooperative work (to learn how to work with others, to undertake commitments, and to ascribe responsibilities); reflective learning (to develop consciousness and metacognition); and other active learning strategies. We also want to place special emphasis on transformative teaching and learning methodologies such as the co-creation of emerging futures. This methodology is based on the idea of doing actions in the present that contribute to the future having the potential to be as we want it to be [37]. Theory U from Otto Scharmer [38] offers an approach and a methodology for personal and institutional change that is based on the following ideas: abandoning preconceptions, working at the systemic level, embracing change, and learning how to co-create transformative futures. It is a methodology widely used in internationally recognized centers working with innovative teaching methodologies to advance towards sustainability, as is the case of the Schumacher College.

## 3. The Design of the *ImpSDGup* Course

The purpose of the *ImpSDGup* course is to enhance the professional development of university teachers in the field of ESD. To this end, the course places special attention to the implementation of the SDGs into the curriculum of university subjects. The literature on ESD training courses and modules for change have been increasing during the last two decades [16,17,35]. Table 1 distinguishes the most interesting and common learning activities found in the literature.

### 3.1. Course Outline

The contents, design, and structure of the *ImpSDGup* course is inspired in the TMTAS model and the main trends in the literature on ESD training courses (as shown in Table 1). The main objectives of the *ImpSDGup* course include:

- Understanding the complexity behind sustainability by identifying the three dimensions of sustainability and how they are interconnected and represented in the SDGs.

- Identifying the main characteristics of the local, regional, and global contexts that affect higher education (HE) and define possible actions for the achievements of the SDGs.
- Broadening the knowledge in sustainability and acquiring ESD competences, teaching methodologies and strategies, and theoretical frameworks in sustainability in HE.
- To apply the contents of the course in reorienting the design of a university subject course towards sustainability.

**Table 1.** A synthesis of stages and learning activities in education for sustainable development (ESD) training courses.

| Main Activities | Description | Based on |
|---|---|---|
| A starting activity | Participants reflect on their own knowledge and competences on sustainability and ESD. | [17] |
| Clarification of the concepts of sustainability and ESD | Group work strategies to understand and share the meaning of sustainability and its principles. | [39] |
| Contextualization | An analysis of the institutional context (individually or in small groups) and the identification of how their disciplines can contribute to the SDGs' agenda. | [8,18] |
| Exemplification of ESD methodologies | Exemplification of how to use sustainability pedagogies and methodologies and how to enrich the learning experience of students with their informal curriculum. | [40,41] |
| Experimentation with interdisciplinarity | To experiment with types of interdisciplinarity and transdisciplinarity by taking students, other academics, as well as citizens' perceptions, demands, and proposals into account and allow them to have a say in the development of their community. | [42] |
| The transformation of current practices and the creation of new proposals | To utilize an ecological course design process that weaves the previous dimensions together to create transformative learning experiences. | [36] |

The course consists of three face-to-face sessions of five hours each and posterior personal work at home that is dedicated to the design of a project on curriculum reorientation towards sustainability.

3.1.1. First Face-to-Face Session: "From the Past to the Present"

Outline: During the first session, the course analyses past and current causes and consequences of unsustainable practices. Participants are introduced to the concepts of sustainable development and ESD, to The United Nations Agenda 2030 and its 17 SDGs, and to the main achievements in sustainability in Spanish higher education institutions. In the last part of the session, teachers are asked to openly reflect on what knowledge, skills, competences, and values they have and need in order to address the challenge of sustainability.

Development: In the first part of the session, participants analyze the Brundtland report definition of sustainable development [43] and the one provided by the Center for Sustainable Systems (University of Michigan) [44]. The facilitator stresses the importance of the three dimensions of sustainability (environmental, social, and economic), the systemic interaction between them, and the complexity of the concept. The following activity is focused on the evolution of ESD, wherein participants discuss the definition made by Martínez [45], who states that ESD is the training for conscious action with the aim of learning to change, also adding that the best strategy for achieving this aim is through the participation in real projects of transformation and change.

The SDGs are also part of the work. A participatory and critical analysis of the 17 SDGs is carried out, placing special attention to SDGs 4 and 8.

The session continues with a "sustainable perspective presentation" where both the facilitator and the participants show how the three dimensions of sustainability are represented in the different disciplines and knowledge domains in which they teach. This fact is fundamental to maximize the

possibilities of success of the course, as all the contributions are necessary to build that systemic and interdisciplinary vision that is essential to advance in sustainable proposals and solutions.

Finally, the session ends up with a mapping exercise to identify the participants' previous knowledge in relation to sustainability and helps the facilitator in identifying those aspects that need more attention during the remaining sessions of the *ImpSDGup* course. Participants form small working groups to reflect on three main elements (that are divided in colors): knowledge (red), capacities and skills (blue), and values (green). The results are presented on the board by theme and color and they help in initiating the final discussion.

### 3.1.2. Second Session: "From the Present to the Future"

Outline: In the second session, the course is focused on ideas from the present and we focus on the ESD competencies we need as university teachers to be able to base our teaching on sustainability [29,34]. Participants analyze the context in which their teaching takes place by means of a strengths, weaknesses, opportunities and threats (SWOT) matrix. The session ends up with a "Y Exercise" [46].

Development: In the second session, participants reflect on ESD competences using the document Education for Sustainable Development Goals [47] and paying attention to in competencies such as anticipation competence, normative competence, strategic competence, collaboration competence, critical thinking competence, self-awareness competence, and integrated problem-solving competence. We also carry out a dynamic to identify which of the competences of teachers in sustainability we have already worked on and which we should train in. To do this, we use the document "Learning for the future: Competences in Education for Sustainable Development" [34], in which the focus of the competences is coherent with the approach to the design of our course, on the basis of the co-creation methodology. In the case of the document, it can be seen that the competencies learning to know, to do, to live together, and to be are presented in three different aspects: the holistic approach (as a representation of inclusive and practical thinking), the change in the environment (viewed from the past, present, and towards the future we wish to build), and those competencies necessary to achieve transformation (in relation to people, pedagogies, and educational systems).

The local context, the institutional context, and the context of the working groups have a decisive influence on the ideas that we finally manage to put into practice. Identifying them, on the one hand, allows us to set ourselves coherent and achievable goals in the short/medium term. However, it also allows us to identify the aspects that need to be transformed in the long term to move forward. Without this dual approach, the possibilities of transforming the environment and the education system are limited. In this way, participants can begin to envision what their own route to transforming the teaching they do may be and make it increasingly sustainable.

The final activity of the second session is the "Y exercise". It asks participants to choose a lesson plan or unit that is currently teaching and identify: the three dimensions of sustainability (environmental, social, and economic) and the knowledge, local issues, perspective skills, and values that already exist in their units and those elements that are missing and that they would like to add.

### 3.1.3. Third Session: "The Future"

Outline: The focus on the third session is on the design of more sustainable common future scenarios, through the contents, teaching methodologies, and values of sustainability that appear in examples of good practices, with various resources and tools thus being explored.

Development: Throughout the third session, participants work on the importance of designing more sustainable future scenarios and on the need to put the principles of sustainability into practice in subject curriculum design. To this end, the TMTAS model (Figure 1) is presented as a guiding framework for the development of their proposals.

### 3.1.4. Autonomous Work

Outline: During the autonomous work, participants are expected to complete their final assessment work, consisting of a proposal of reorientation of the program and/or curriculum of a subject course that they deliver at their universities. The proposal must follow the TMTAS model and reflect on all the contents of the course. This project is compulsory, and participants must submit it in order to obtain the attendance certificate.

## 4. Methodology

### 4.1. Study Context and Sample

The *ImpSDGup* course is due to the commitment of the Office of Cooperation for Sustainable Development (OCDS) at University Jaume I (UJI) and the Ministry of Transparency, Social Responsibility, Participation and Cooperation of the Community of Valencia's Government, which provides resources to facilitate the professional development of university teachers, understanding that this is a fundamental aspect for the transformation of both teaching and the whole institution towards a more sustainable framework.

The *ImpSDGup* course has been delivered three times at the UJI from 2017 to 2019, always using the same methodology. Until the 2019 edition, a total of 55 university teachers from different disciplines had participated: 21 in 2017, 17 in 2018, and 17 in 2019. The Centre for Postgraduate Studies and Lifelong Learning (CPSLL), which is part of the UJI Studies Office, is the responsible party for inviting and encouraging the UJI members of the teaching staff to participate in the course. It is worth stating that the CPSLL does not carry out a selection process with the candidates because one of the demands of the course is to mix university teachers from different fields of knowledge and with different ESD backgrounds.

### 4.2. Research Design and Data Analysis

We considered a qualitative methodology as the most appropriate approach for this study because of the nature of the research problem—to understand how university teachers reorient their curriculum towards sustainability. Qualitative analysis has the quality of being flexible and adapting to the reality being researched. Furthermore, it provides depth to the analysis of data, dispersion, richness of interpretation, details, and unique experiences [48,49]. The approach for qualitative data analysis that we are using in this research is content analysis. For Krippendorp [50], content analysis is a research technique for making replicable and valid inferences from texts to the contexts of their use. It can be undertaken with any written material and, in our case, the instrument of data collection was the program of one subject course that participants elaborated after the sessions of their ESD training. Participants were asked to develop a reorientation proposal towards sustainability with the following requirements or parts: the title, the definition of the main objectives, a clarification of competences, an explanation of the ESD methodologies to be implemented and how this will be done, a detailed description of learning activities (and the associated teaching and learning resources), and a final reflection comparing the new design with the previous one.

We took into account the steps for the whole process of content analysis suggested by Cohen et al. [48] and we reformulated them as follows:

*Step 1: The definition of the research aim and the questions to be addressed*. The purpose of our study was to investigate what ESD transformations participants of the *ImpSDGup* course implement in the design of the curricula and programs of their course subjects. To this end, we asked the following questions: What contents, ESD methodologies, and competences treated during the training course are present in their proposals? What are the main transformative actions?

*Step 2: The definition of the population, the sample, and the context of generation of the documents used in the research.* In this study, we examined the final work of the 55 university teachers that participated in the *ImpSDGup* course. In this case, the population and the sample were the same.

*Step 3: Definition of the units of analysis.* We identified and selected the basic meaning units from part of the data contained in the programs. We focused our attention to course objectives, the title of course themes, the main statement and description of the activities, the competences, and the statements of the assessment criteria.

*Step 4: Codes and categories for the analysis.* Codes are the basic analytic units that have been identified and ascribed to basic meaning units. Coded segments have been grouped into categories. For example, SDG4 is considered a code that is part of the SDGs category. The categories, in turn, have been classified into three main themes: contents, ESD methodologies, and competences in sustainability. The themes were chosen from the TMTAS framework.

*Step 5: Other emergent findings.* The writing of the final summary was structured following the three main themes and adding and emergent fourth theme: the relationship among themes.

### 4.3. Ethics

This research was accomplished with the following ethical principles: participants of the course were informed at the beginning of the first session about the intention to use their final work in a research study on curriculum reorientation towards sustainability; the aim of the research was briefly explained; through a consent form, participants gave permission with explicit authorizations for the examination of their documentation; the authors of the research maintained the confidentiality of participants and data were only related to disciplines; the research implied a low risk for the people involved; authors conducted the research following the principles of fidelity, responsibility, integrity, and respect for participants' work.

## 5. Results and Discussion

The three editions of the course counted with a representation of professionals from 25 different disciplines within the university (see Figure 2). Tourism was the discipline with the majority of participants. It was followed by engineering, mathematics, psychology, and chemistry. If we classify these disciplines by their relationship with environmental, social, or economic studies, we can see how the social branch was the most represented, followed by the economic branch (see Figure 3). The low participation of university teachers from environmental subjects should be further investigated in the future to understand why they do not perceive this training as something fundamental for their professional development.

The variety of disciplines was proven to be convenient for the delivery of the course as it helped in the application of the co-creation methodology that encourages the sharing of knowledge, expertise, and experiences of the participants, benefitting them with complementary perspectives and approaches to sustainability. As a result, this can strengthen the knowledge of the dimensions of sustainability that some participants are not familiar with due to the highly specialization of academia today.

Next, we present the results of the study according to the three main elements of the TMTAS framework: the contents of the subject, the ESD methodologies, and competences for sustainability. A fourth section was added in order to integrate other emerging results.

### 5.1. Contents of the Subject

This theme refers to the contents of the subject courses that university teachers deliver at UJI. Data were obtained mainly through the learning objectives and the statements of the units of the course. The findings regarding this theme are structured around three categories: SDGs, sustainability principles and concepts, and sustainable futures.

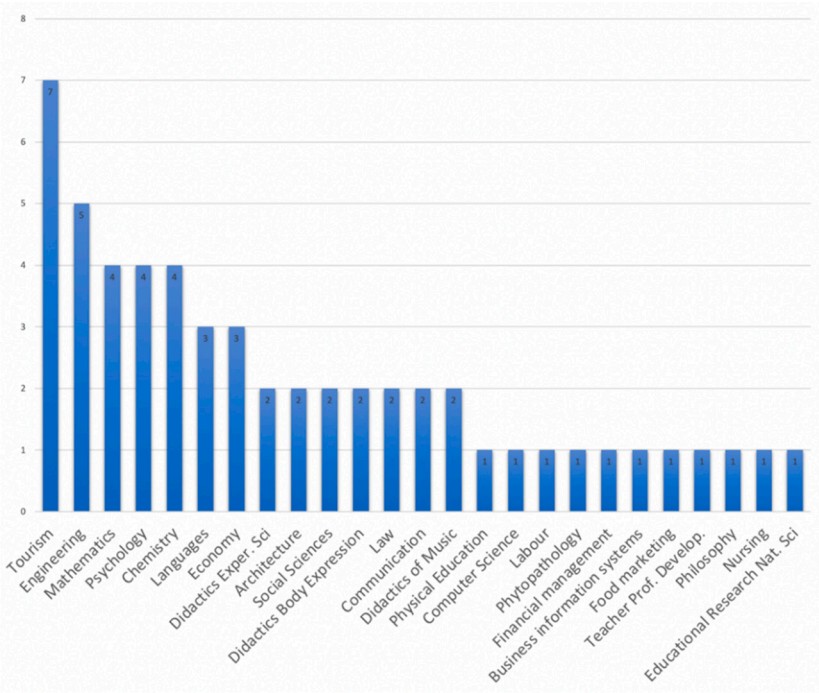

**Figure 2.** Number of participants by disciplines.

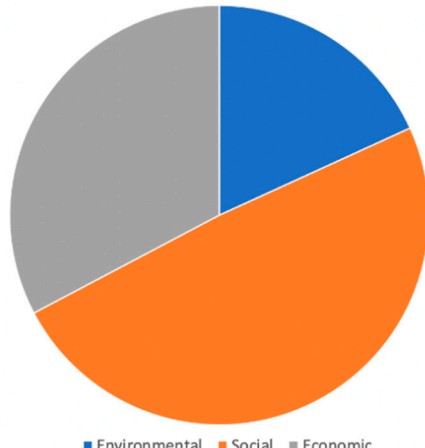

■ Environmental ■ Social ■ Economic

**Figure 3.** Disciplines of participants in relation to environmental, social, or economic studies.

5.1.1. SDGs

The comparison of new and previous learning objectives demonstrate that university teachers have found Agenda 2030 to be something necessary and of great importance today. They found that the list of SDGs was a useful framework to take into account at the time of selecting the contents of their subjects. Firstly, university teachers started to realize that some of the SDGs were already part of their teaching, and thus they simply identified and mentioned them. Secondly, they explored Agenda 2030 and incorporated new SDGs into their programs. As a result, an important number of SDGs are now present in the new programs of the subjects (see Table 2). Each subject deals with a minimum of five SDGs, with being SDG13, SDG8, and SDG17 being the most repeated ones. The introductory text of the subject of Didactics of Music is an example of explanation of how music is related to some of the SDGs:

> " . . . *music can reach where words cannot, but it can also raise awareness and influence the conscience of people from many and diverse transversal themes, which can range from noise pollution, to the misuse of recycling (SDG12) or also from research projects to combat poverty (SDG1) or improve the*

*quality of life of people (SDG3). There are many songs that we can find through the national and international repertoire that are intended to raise awareness among children and adults about the environment, healthy habits for health and quality of life (SDG3), gender equality (SDG5), and so on. For this reason, it is important that this discipline, from a teaching point of view, should become aware of the great and serious global changes, in order to educate in ethics and conscience, in values and for a better society".* (DidMusic-SDGs-5)

**Table 2.** Sustainable development goals (SDGs) found in the content of subjects.

| Subjects/Disciplines | SDGs |
|---|---|
| Languages | 4, 5, 12, 16, 17 |
| Tourism | 6, 8, 11, 13, 14 |
| Engineering | 7, 8, 9, 11, 12, 13 |
| Didactics of experimental sciences | 3, 4, 5, 6, 10, 13, 14, 15, 16 |
| Architecture | 1, 3, 7, 8, 9, 11, 12, 13 |
| Social sciences | 1, 4, 5, 8, 10, 13, 16, 17 |
| Mathematics | 8, 9, 12, 13, 14, 15 |
| Didactics of body expression | 3, 4, 5, 10, 13, 16, 17 |
| Psychology | 3, 4, 5, 8, 10, 12, 13, 16, 17 |
| Chemistry | 2, 3, 4, 6, 8, 9, 12, 13, 17 |
| Law | 1, 3, 8, 10, 12, 13, 16 17 |
| Physical education | 3, 4, 5, 8, 10, 13, 16 |
| Computer science | 1, 2, 3, 4, 5, 8, 9, 10, 11, 13, 17 |
| Communication | 1, 2, 3, 4, 5, 8, 10, 13, 16, 17 |
| Occupation–labor | 3, 6, 7, 8, 9, 11, 13, 14, 15, 17 |
| Phytopathology | 2, 3, 5, 8, 9, 12, 13, 17 |
| Financial management | 3, 8, 9,12, 13, 17 |
| Didactics of music | 1, 3, 4, 5, 10, 11, 13, 14, 15, 16, 17 |
| Business information systems | 1, 2, 7, 8, 9, 11, 12, 13, 17 |
| Food marketing | 1, 2, 3, 6, 7, 8, 9, 12, 13, 16 |
| Teacher professional development | 1, 3, 4, 5, 8, 10, 12, 13, 16, 17 |
| Economy | 1, 2, 7, 8, 9, 10, 12, 13, 16, 17 |
| Philosophy | 1, 3, 4, 5, 8, 10, 11, 12, 13, 16, 17 |
| Nursing | 2, 3, 4, 5, 6, 8, 10, 11, 13, 16, 17 |
| Educational research in the natural sciences | 3, 4, 5, 6. 7, 12, 13, 14, 15, 17 |

In mechanical engineering, the department was already working on SDG7 through sustainable energy, but in the new program they included SDG9 (industry innovation), SDG11 (sustainable cities and communities), and SDG13 (to contribute to climate change through measures of $CO_2$ reduction). Another subject that demonstrates its wide potential in contributing to sustainability is didactics of physical education, which aims to "empower students to be able to apply the perspectives of the SDGs health and welfare, quality education, gender equality, reduction of inequalities and partnerships to the design of their classes in order to achieve the objectives". It is particularly positive that SDG5 (gender equality) is considered essential for the subject because it identifies that physical education is full of gender preconceptions such as "girls can't do that", "boys are stronger", and "girls don't play

football", among many others. The elimination of these types of stereotypes would be easier if there were critical thinking among students and if SDG5 was incorporated in most of the studies.

In some other cases, the SDGs were integrated in a generic way:

*"To guide the future engineer to adopt transversal action strategies in the field of phytopathology in which the greatest number of SDGs are taken into account".* (Phytopathology-SDGs-1)

On the other hand, we must point out that SDG1, SDG11, and SDG16 were the least implemented goals. For example, participants recognize the importance of poverty, but failed to address it in their subject courses.

### 5.1.2. Sustainability Principles and Concepts

Most of the sustainability key concepts and principles appear throughout the texts that have been analyzed. In a general sense, there has been an evolution in the approach of the conceptualization of the contents of the subject, moving from a traditional and linear approach, towards a more holistic vision, implying a reflection on what it means to apply the principles of sustainability to teaching. For example, within the course on phytopathology, we can identify significant changes such as the addition of the scale analysis and the need for contextualization:

*"To know the local and global framework in which phytopathology is inserted, and the direct application of what has been learned in the classroom".* (Phytopathology-SustP-2)

*"To know the different known diseases caused by fungi, bacteria and viruses, contextualized in the environment".* (Phytopathology-SustP-4)

*"To contextualize the study of plant health in the framework of world health, with reference to the state of global health".* (Phytopathology-SustP-7)

University teachers also developed the ability to think in terms of the three dimensions of sustainability when defining the objectives of their courses. For example, in the case of the course "mathematical modelling and optimization in the field of engineering", we appreciated a qualitative improvement—the initial objective was transformed into a new objective that incorporated models for analyzing the environmental, social, and economic aspects of unsustainability and for generating solutions to mitigate climate change, generate jobs, and minimize costs. In the subject "teacher professional development", it was stated that "we will identify the environmental, social and economic components of the SDGs and define areas of action" (TeaProfDev-SustP-2). These examples denote an integrative approach to sustainability.

### 5.1.3. Sustainable Futures

This category refers to the use of new proposals and good practices on sustainability as content for teaching and learning. Those practices act as sources of inspiration that guide us in the imagination and creation of sustainable future scenarios. For example, the model of the economy of the common good is studied through a case study in the subject of "psychology". The transition network and the circular economy is used in the subject "English language", where it plans "to contribute from communication to the dissemination of experiences that are already being implemented around the world to promote a culture of solidarity, focused on supporting each other, as groups or as larger communities" (Lan-SusFut-3). The knowledge and analysis of models, experiences, and good practices is critical in the professional development of university teachers because they provide tools and practical examples of how ideas are transformed into reality. If university teachers are confident about their knowledge on sustainability, their impact on students' learning and empowerment will be higher.

### 5.2. Teaching Methodologies and Learning Activities

The analysis of this theme is divided in two categories: active methodologies and transformative methodologies.

### 5.2.1. Active Methodologies

The need for an educational reform that bets on the use of active methodologies and the diversification of class strategies has been stressed for a long time, since there is a wide range of possibilities that allow the work of different ESD competences [51–53]. In the *ImpSDGup* course, we try to be consistent with theory, and we deliver our sessions using active methodologies. It seems that this approach encourages university teachers to replicate them—in each of the final projects, we found the implementation of ESD methodologies such as challenge-based, problem-based, and project-based learning; service-learning; debates; simulation; role-plays; and critical evaluation of case studies. For example, in the subject "strategic marketing decisions", the teaching methodologies of challenge-based learning and project-based learning are introduced. The former does so through a proposal of "deconstruction of marketing: how to redefine it in the context of the new economies" (Mark-TeachMet-2), which also makes the student familiar with these economies and encourages them to rethink the discipline in a different context. In the second case, they must "deliver a project to transform the trend known as fast fashion into its sustainable version which is slow fashion" (Mark-TeachMet-4). Once again, they must familiarize themselves with a proposal to move towards sustainability in one of the most polluting industries that exists. In another example, the study of the role of women in psychology was reduced to discovering their contributions through knowledge of their work, which in the final work proposes a gender analysis that takes into account social, cultural, and economic factors and how they have determined their influence on the history of psychology. Moreover, in the field of psychology and from the perspective of psychosocial intervention, special emphasis began to be placed on storytelling, which makes it possible to take advantage of the lessons of previous experiences so as not to repeat the same mistakes and to identify patterns of behavior. The implementation of the first U laboratory was proposed on the basis of the transforming methodology Theory U [38], which is theoretically very appropriate for these scenarios in which the fundamental objective of intervention is transformation and not the specific solution of a problem.

The subject "culture and development" started to use cooperative work, the flipped classroom methodology, and techniques to stimulate participation such as the puzzle, with the aim of working on cosmovisions of different peoples and/or social groups (in this case indigenous peoples, pantheists, naturists, theists, and animists). The cosmovision determines how we see the world and therefore how we act, with this analysis being fundamental for sustainability and transition processes as it entails intercultural dialogue.

In business studies, the subject courses "analysis and formulation of business strategies" and "strategic management" implemented a diversification of the type of companies that formerly were analyzed in order to connect with other organizational realities whose objectives cover non-profit actions: " . . . a greater number of problems will be introduced that not only seek to maximize the company's economic results, but also to reduce waste and other environmental costs, improve the working conditions of employees (e.g., hours of training, gender equality at work) and increasing the long-term resource efficiency of the company (e.g., reduction of waste) . . . we will use methodologies such as project-based learning, role playing and cooperative work for the development of databases using Microsoft Access . . . " (BusSt-TeachMet-3). These two courses also incorporated critical and future thinking through the formulation of sustainable alternatives.

5.2.2. Transformative Methodologies

Transformative methodologies promote a sustainable citizenship that is characterized by its commitment with the transformation of society, entailing responsibility, critical thinking, and a continuous socio-educative process.

> "*change in a frame of reference encompasses cognitive, conative, and emotional components, and is composed of two dimensions: habits of mind and a point of view. Habits of mind are broad, abstract, orienting, habitual ways of thinking, feeling, and acting influenced by assumptions that constitute a set of codes. These codes may be cultural, social, educational, economic, political, or psychological. Habits of mind become articulated in a specific point of view which highlight the constellation of belief, value judgment, attitude, and feeling that shapes a particular interpretation. On their behalf, all these ideas let to know how deep the challenge of implementing the transformative learning is that change in behaviour and in cosmovisions needs a learning process of major complexity in which meaning perspectives are transformed*". (Mezirow [32] pointed out, p. 5)

The use of the co-creation as a transformative methodology during the *ImpSDGup* course transmitted hope and had a remarkable impact on university teachers. For example, in the subject "Spanish language", the teacher proposed "to co-create a seminar to analyze the role of language in the creation of meanings and demonstrate the relationship between the vocabulary of different historical moments and the magnitude of the type of gender discrimination. We will take a journey through the last 100 years of history, going through analyzing the current situation until we come to visualize how we would like the gender relationship to be in the future and what type of language could favor it ... " (SpaLan-TransMet-1). Another example is that of the "strategic marketing" subject, wherein the co-creation of emerging futures is included in its syllabus, being considered as "[a]n active, creative and social process, based on collaboration between producers and users that is initiated by the firm to generate value for customers, in order to develop their products with sustainability criteria and begin to circularize the life cycle of the company. This means working with the end users of your product or service to exchange knowledge and resources, in order to offer a personalized and redirected experience when analyzing the environmental, social and economic impact of the product, through its evolution over time, to build a version of the future in accordance with the philosophy of sustainability" (StraMar-TransMet-5).

*5.3. Competences for Sustainability*

SD competences have been worked extensively by several authors [28,29,34,51–53]. Systems thinking, normative capacity, strategic capacity, competence in anticipation, critical thinking, self-awareness, and problem-solving are some of the main components of the ESD competences that have been selected by participants to be included in their proposals. Whether or not these skills become a reality will depend on both the content and, in particular, the teaching methodologies, strategies, and resources used, and moreover, on the roles adopted by teachers and stimulated in students, on the leadership capacity of the teacher, and on the coherence between discourse and practice at different levels.

Skills such system thinking, critical thinking, and the design of innovative proposals and solutions for sustainability are selected by a wide range of university teachers in their new proposals. The subject "planning and integrated management of tourism development" has shown how to implement systems thinking within an accurate understanding of the SDG6 and SDG 11 and some of their targets:

> " ... *the purpose of the exercise is to analyze, from a systemic perspective, the physical-natural environment and the demography-socio-economy. Thus, in the analysis of the physical-natural subsystem, based on the principles of Sustainable Development, SDG 6 will be highlighted. This is justified by the fact that water resources are essential for tourism activity, in the analysis of the recycling and reuse of wastewater (Target 6.3); and in the analysis of the degree of protection of*

*water-related ecosystems, in the case of aquifers (Target 6.5). Regarding SDG 11. Ensure that cities and human settlements are inclusive, safe, resilient and sustainable; the current level of protection of the natural heritage will be analyzed (Target 11.4). From a socio-economic perspective, as in the previous one, based on the principles of Sustainable Development, SDG 11 will include sustainable cities and communities in its analysis, since the development of tourism activity in the receiving area leads to changes in land use and transforms the territory with the appearance of new human settlements, which are translated into tourism destinations".* (PlanTour-Compet-3)

This example makes clear a fact that occurs in many other cases, which is the wide versatility of the SDGs as a route for developing competences for sustainability.

Innovation is promoted through subjects such as "moral philosophy" and "chemistry of materials":

*To elaborate intervention plans based on the SDGs using them as a key instrument in the development of social innovation processes . . . .* (MorPhi-Compet-4)

*To design innovative proposals for new ecological packaging designs, taking into account their impact on the economy and the environment.* (CheMat-Compet-1)

*To know and elaborate proposals from the point of view of the circular economy of packaging: Concepts of Optimization, Reuse and Valorization of packaging.* (CheMat-Compet-3)

Another special case that implies the creation of solutions together with critical analysis is the subject of "mathematics", where students are trained

*[t]o apply mathematical modelling appropriately to the solution of sustainability and sustainable development challenges and to integrate and critically analyze the result of using mathematical models in the definition of solutions and strategies for sustainability and development.* (Maths-Compet-2)

*5.4. Other Emergent Findings*

We believe that the *ImpSDGup* course has the potential to increase its impact in the entire UJI institution, as many actions were initiated thanks to the course. Evidence of this includes the coordinator of the third year's project of architecture studies stating, " . . . *by taking advantage of the course, a step forward has been taken. This makes use of all those aspects that, in some way, were already consistent with the integration of the SDGs into teaching, and it has been expanded in accordance with the new knowledge acquired . . . ".* Thus the 17 SDGs became the central axis of the project, from which the students must carry out projects that contribute to the fulfilment of those goals (and specifically, SDG11 "sustainable cities and communities" became mandatory). Furthermore, the coordinator expressed a very high level of commitment, " . . . *[a]ware of the context in which we find ourselves and of the responsibility in the training of future Technical Architects, the TAG wants to join this transformation process and therefore assumes the commitment to adapt its curriculum by integrating sustainability as a transversal objective from the ecological, economic and social approaches. Likewise, it is committed to promoting the inclusion of specific sustainable objectives in each of the subjects . . . ".* The associate dean of the Faculty of Education, in the 2018 edition, created a new seminar on educational innovation (SPIE) under the name of "Harmonies of Environments for the SDG", with which a group of professors launched a proposal for joint work, with the ideal of proposing an interdisciplinary space for the degrees of the University Jaume I. This led to the work entitled "*Sustainable development in teacher training. Cultural materials and concepts applied in educational environments*", presented at the conference "Advances in Technologies, Innovation and Challenges in Higher Education (ATIDES)". The same group of SPIE teachers organized and carried out the first International Teachers' Congress: *Interdisciplinarity and Sustainability through the Arts*, which was held from 6 to 8 November of 2019 at the UJI. Finally, another professor participating in the *ImpSDGup* course prepared the document "*Open letter to the University Jaume I Governing Board for the declaration of the state of climate emergency*", which was displayed on the interactive platform of the

same name with the aim of mobilizing all the actors of the UJI for the purpose described. All these actions let us think of the great impact the course had on its participants and on the whole institution. Those actions encourage us to continue with the course and its continuous improvement.

## 6. Conclusions

The first aim of this paper was to justify the design and delivery of the *ImpSDGup* course as a teacher training proposal for the integration of sustainability and ESD in university subject courses. We focused on two main questions: How can we reorient our university courses and embed ESD? How can university teachers be trained in this field? The revision of the literature led to the development of the TMTAS framework, which proposes that ESD university teacher training should focus on contents of sustainability (mainly through the 2030 Agenda), theoretical approaches, and ESD active and transformative methodologies. The design of the *ImpSDGup* course is structured around the transformation of the curricula of university subject courses. In order to achieve this objective, participants of the course are exposed to the following main learning activities: the cocreation of the meaning of concepts such as sustainability and ESD, an exploration of the SDGs, a revision of how the content of their subjects is related to sustainability and the SDGs, the introduction to active and transformative methodologies in teaching and learning, and a redesign of the current programs and curricula of their subjects.

The second aim of the paper was to investigate what ESD transformations participants of the *ImpSDGup* course implemented in the design of their curricula and programs of their course subjects. We can state that this training helped participants in transforming their subject curriculum towards sustainability to a certain extent. The analysis of their work proved an understanding of the ESD approach of the course and an effective use of it in the redesign of their subjects. In terms of contents, university teachers at UJI demonstrated the inclusion of sustainability when defining course objectives and addressed most of the SDGs in their course units. In the objectives of the courses, we were able to appreciate the design on the basis of a holistic perspective, reflecting the indissoluble relationship between the three dimensions of sustainability. We believe that this happened as a result of the diversity of participants' background and the course design, which enabled a meeting space where participants (as experts from different fields) exchanged their perspectives and views on sustainability and were exposed to other subjects' knowledge and to interdisciplinary work. Most of the university teachers have embedded active methodologies in their teaching such as problem-based learning, challenge-based learning, and service learning. The teaching and learning activities that were proposed by participants showed their coherence with the objectives they set, and there was an understanding of the need to put students in an active role with regard to learning and to take advantage of the potential offered by technology. In contrast, only few of them felt ready to introduce transformative methodologies. Participants manifested in their writings that they need more training in these methodologies in order to deliver real transformative learning experiences. The competences that were listed in their courses were also aligned with the SDGs and ESD. Most of the programs included systems thinking, critical thinking, and problem-solving strategies. Storytelling was widely used to work on values and emotions related to sustainability. All these findings demonstrate that university teachers managed to improve their professional competences in reorienting the curricula towards sustainability, and now they feel more confident.

This study presents some limitations. The first limitation is that the analyzed data only looked at the curricula and/or programs of the subjects' courses and not to their implementation. We now know something about university teachers' "declarations of intents", but we do not have data from real practice. For this reason, further research must combine different sources of data, including programs, reflection on practice, and evidence from students' learning, to mention a few. A second limitation is that the study focused on single course subjects and personal changes. We consider that the reorientation of subjects' curricula towards sustainability is necessary in higher education, but if it becomes the center of attention, it will prevent us from discovering and appreciating other

institutional transformations that are necessary for the transition towards sustainability. This takes us to the third limitation—interdisciplinarity has been introduced within some subjects' courses but nothing is reported about proposals that take interdisciplinarity and transdisciplinarity beyond the current organization and institutional structures. These limitations have implications for the *ImpSDGup* course—if trainers of the course pretend to fill those gaps in the future and provide a next level of transformation in higher education, they will need to reformulate some aspects of the focus of its training and recruit participants from the leadership of the institution.

**Author Contributions:** The two authors have contributed equally in the research and in the creation of the manuscript. They have read and agree to the published version. All authors have read and agreed to the published version of the manuscript.

**Funding:** This research received no external funding.

**Acknowledgments:** We would like to thank to all UJI university teachers that participated in the *ImpSDGup* course for facilitating the use of their proposals in this research.

**Conflicts of Interest:** The authors declare no conflict of interest.

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
