# Peer review of "Implementation of SDGs in University Teaching: A Course for Professional Development of Teachers in Education for Sustainability for a Transformative Action"

_sustainability, doi:10.3390/su12198267_

Round 1

Reviewer 1 Report

The topic of the paper is very up-to-date and corresponds to the topic of the journal itself. The aim of the paper is to enhance the skills of university teachers in the direction of sustainable development, which coincides with the goals of the United Nations Agenda 2030. To achieve this, the authors of this paper have developed a new course called ImpSDGup based on the Model of Training for Sustainable Action and Transformation (MFATS). The advantage of this model is in the active and transforming methodology. The ImpSDGup course was applied to the Jaume I University (UJI) in the period from 2017 to 2019 and included a total of 55 professors, from different fields, in the mentioned period. The paper further analyzes the success of the ImpSDGup course itself on the basis of three categories: objective of the subjects, teaching methodologies and class activities, and competencies in relation to the other categories.

The paper approaches the topic of Education for Sustainable Development very broadly. However, I am of the opinion that in some parts the work should focus more on the model that has been developed and its results, which would mean that the work should be shortened and unimportant parts should be eliminated. The key words should be more specific, and not in the form of whole phrases as it written in the paper. Regarding to the use of the English language, the paper is understandable, however, does not explain enough abbreviations that are in it. Namely, each abbreviation must be explained when it is used for the first time, which is not the case in the paper. Also, some of them like ODS (Objectives of Sustainable Development) are not used as such in the original format but SDGs (Sustainable Development Goals) are used. In addition, reference 15. contains non-English words. Figure 1 should be in better resolution. Finally, I am of the opinion that the hypotheses of the paper should be more clearly emphasized in the text itself.

Author Response

Please, take a look to the documment attached.

Kind regards,

The authors

Reviewer 2 Report

Overall impression

I agree with the authors that sustainability is a very important issue for university education, in both the curriculum and in the practices and policies that are enacted locally. This focus to the study is commendable and certainly needed in the field. I thought that the diagram was most promising and there were ideas throughout the manuscript that are worth developing.

However, this writing is problematic organisationally and in terms of its academic English. There is a lack of a consistent structure to the material, and material that is not presented in appropriate sections. The article needs a through rewrite in order to structure the material in an appropriate sequence. Moreover, the academic English is truly incomprehensible at times. Your language lacked precision and specificity. At times I was struggling to understand what ideas you were referring to. This manuscript seemed to me to be like an early draft, rather than a precise, well organised and polished manuscript ready for review.

Finally, there is methodological and theoretical confusion in this writing.  I was not sure what you were exactly researching and with what methods. Only towards the end of the article did some of this become even a little clearer. I am also uneasy about the nature of the research. Just evaluating a teaching program hardly constitutes original research worthy of a top journal. The authors need to consider what this research is really about and what is its original contribution to the field.

Specific feedback

  1. “The words unsustainability and uncertainty leave less and less room for doubt. When talking about sustainability just a few years ago, most people associated it with an event of the long-term future.” This is a statement of exaggeration and not supported by the evidence of research. The authors need to take care with such blanket claims.
  2. “despite the increase in the content conveyed on sustainability, ecological, economic, and social problems continue to grow.” Are the writers suggesting a causal link here? Again, problematic.
  3. “trained to exercise the most diverse professions and, therefore, to design this sustainable system.” Very vague and generalised. What do the writers mean by ‘sustainable system’?
  4. “The need to make the university curriculum more sustainable”. Do the writers mean more focus on sustainability? There is ambiguity here.
  5. “evaluating its scope to favors curricular sustainability”. This does not make sense.
  6. There seems to be a confusion between sustainability as a curriculum initiative in higher education and the policies around sustainability as practices within educational institutions. The authors must clarify what they are pointing to. Is it about sustainable development or content about sustainability in the teaching and learning programs within university? This is not clear in the manuscript.
  1. “it is expressed and exemplified how training models committed to the perspective of complexity should reflect a systemic, dialogical and hologrammatic approach”. You have introduced a whole lot of concepts here that are not explained and lack supporting citations. For instance, complexity theory is a whole field in itself and no clear understanding of this idea is offered. In the material that follows thee is an attempt to unpack these ideas. But the language is rather lacking in clarity, and, again, no citations or references to the field are made.
  2. “ESD relates to the essence of teaching and learning and should not be considered as an annex to the current curriculum (11). It is essential that subjects are designed with a focus on ESD by making values, skills, content, teaching methodologies and teaching and learning resources and tools coherent and closely coordinated so that they reinforce each other.” This has much better clarity and it is where the paper should be focused. Once again there few citations offered.
  3. “Kolb's theory of the empirical learning cycle” Should this not be ‘experiential’? There seem to be little understanding of Kolb and the relation of his theory of learning to ESD.
  4. What follows are more theories that are supposed to link to ESD but again these are unconvincing and not properly explained. I suggest that the authors are trying to so too much in this manuscript without doing anything really well.
  5. I found the theoretical framework section fragmented with multiple unconnected use of theories. Seems like a set of discussion notes rather than a cohesive presentation of a framework. The authors have much work to do.
  6. Really the material under Methods about context would have been in the introduction to orientate the reader what this study is actually about. Context is vital but up to this point I had no idea what this study is actually researching.
  7. “Because this study is about the impact of a certain vision of teacher training in HE”. Only now is the specific context of the research actually identified.
  8. “The aim is to study the results of a learning process applied to a real situation such as the design of some of the subjects taught and the qualitative perspective has the quality of being flexible and adapting to the reality being researched. Furthermore, it provides depth to the analysis of data, dispersion, richness of interpretation, details, and unique experiences (16).” This aim should have been identified in the introduction and then reiterated later on. The aim lacks specificity and the language is imprecise and not targeted. The authors need a lot of work in regard to the clarity of their language.
  9. “The interpretative paradigm is considered as interpretative-symbolic, qualitative, naturalistic, humanistic, and phenomenological.” Really? Each one of these is a discrete approach to epistemology. Which one are you actually choosing and where are the citations?
  10. How does a “validation study” sit with these quite interpretive approaches to epistemology?
  11. “evaluated not only by the participants, but also by the researchers, seeking to constantly improve the training proposals.” I wonder whether evaluating a university program actually constitutes proper research. Surely research is about deeper issue, and in this case the constitution of sustainability as a pedagogical practice in the work of university teachers.
  12. In the section on the analysis tool (“Figure 1. Training Model in Transforming Action for Sustainability (TMTAS)” there is much promise and most interesting. I found it difficult to read the text in the diagram, and the diagram itself is very dense. But still, many worthwhile ideas.
  13. “The theoretical approaches respond to the recognition of needs such as the need to transform the current patterns of production and consumption, and the need to move from an anthropocentric to an ecocentric worldview, seen as the prevalence of non-materialistic values such as compassion, imagination, creativity and ethics.” I see that you are attempting to bring some of the values, practices and ideas about sustainability as a measure of a program. This is worthwhile but your tool lacks precision and there are so many ideas that I wonder how you can do a full analysis. Also, and again, no citations.
  14. The “Categories of analysis” were, as I suspected, far too generalised and lacking precision as categories of analysis.
  15. Where is the description of your methods of analysis?
  16. “Description of the work done during the three face-to-face sessions of the ImpSDGup”. Is this your research or merely a description of the UN material? I am not sure what you are trying to do here. Who are the participants in the face-to-face? I am very unsure of what this material is.
  17. You had descriptions of three face-to-face sessions. With whom? Is this a set of teaching events? What went on in the sessions? There is content but not actually what happened and how they were received by participants. Only at the end of the section do you explain that this is a course.
  18. The results section was barely readable and there are so many issues with the clarity of the language that I am not going to list there here. This section lacks precise and logical laying out of your findings. They might have been numbered or put as sub-headings to assist the reader. So many ideas and references in the material were jumbled together that it was difficult to find your way as a reader through this section. This seems like a very early draft.
  19. The conclusion was equally fragmented and contained material that should have been in earlier sections. A conclusion is really a synthesis—it is a pulling together of what you have found and presenting it to the field, reinforcing its originality and importance. There is also a need to consider the limitations of the study. There is none of this type of material.

Author Response

Please, take a look to the document attached.

Kind regards,

The authors

Reviewer 3 Report

Dear authors,
congratulations on your innovative research. Thank you as well for pointing out this important topic.
I have no suggestions when it comes to the paper itself.

However, I believe that it must be proofread. Therefore, it will be improved in terms of style.

Author Response

(The authors gave the same response as above.)

Reviewer 4 Report

This is an interesting manuscript which deals with ESD in higher education and presents the design and results of a training course. It appear that the methodology is very careful followed. However, the presentation of the manuscript is makes very difficult to follow.. Please make your passages more coherent by(a) connecting the short paragraphs, (b) consider in which section you should present the MFTAS model. The results section is well-written, for instance.

Please have a look at my comments on the pdf.

The manuscript should be proof-read for English. It is now difficult to read it.

Author Response

(The authors gave the same response as above.)

Round 2

Reviewer 2 Report

Overall Impression

This revision of the article is much clearer to read and has greater focus and clarity. I commend the authors on the changes made and, on their response, back to me in which attention to each of the points was delivered. This article has a lot of promise and I am looking forward to the final version.

However, there are still some issues that the writers should attend to. The most important one is the tone of the article. Research should not be about promoting an idea or a program but investigating it. There needs to be a level of dispassionate inquiry and then in the conclusion some more expansive suggestions can be made. Second, the conclusion is not good and lacks the usual conventions for a conclusion. Please rewrite and use the conclusion to give the reader a sense of what this research contributes.

Specific feedback

  1. “In this section we try to explore the following two questions” Try to avoid language such as “Try”. It gives the impression of doubt to the reader.
  2. “another where partial transition takes to a complete transformation of the institution”. Do you mean leads to? Not clear.
  3. “Since the mid-nineties, universities initiated to embed sustainability and ESD in their curricula and, although we can acknowledge a considerable progress in the field, some of the needs to be met  are [21]: the creation of institutional guidelines for sustainability; academic engagement and training  in sustainability; curriculum reorientation and innovation; the reduction of universities carbon  footprint; the integration of teaching, research, management and community engagement. This needs to be rewritten and the sentence structure simplified. It is difficult to follow.
  4. In lines 112-151, you use dashes to indicate each of the points. Please use numbers instead.
  5. Inspiring active engagement in the present (which is crucial today because of the emergency of the SDGs); (c) Exploring alternative futures: ESD should lead to positive futures for people and nature, rather than that just reduce harm [34].” I find the language in this section quite emotive and trying to lead the reader to a point of view about the issue rather than more dispassionately describing a program for evaluation. Be careful with the tone of your language throughout.
  6. We need to learn from the principles of living systems and through the philosophy and practice of permaculture design and biomimetics, and to be trained in methods of design thinking inspired in ecology and in the latest advances in anthropology, psychology and socio-political economics.” You are preaching at the reader. Think does not have a place in academic writing. Please evaluate your whole article with attention to the tone of some of the language.
  7. In Section 3.2 Course outline, there is an overuse of bullet points. This seems like a set of notes. Remove the bullet points and use sub-heads to delineate each point.
  8. In the results and discussion section the numbers of participants for each of the discipline areas should have been reported separately in the body of the document, and the text and numbering of Figure 2 is too small.
  9. “The SDGs have resulted to be a clear guide of current urgent issues and teachers have made a great effort in identifying which ones are related to their subject domain.” This sentence does not make sense. Needs to be rewritten.
  10. There are spelling and punctuation errors. Please attend to these throughout the document.
  11. May I suggest to you that what you are doing in this paper is not an interpretive paradigm but content analysis (oriented to document analysis). An interpretive paradigm goes much deeper into the understandings about sustainability (and thus requires different methods), whereas what you are doing is looking at the course content that is related to sustainability. Perhaps you should rewrite this to reflect this difference.
  12. “Mezirow (32) pointed out that change in behaviour and in cosmovisions needs a learning process of major complexity in which meaning perspectives are transformed.” This is a very simplistic understanding of Mezirow’s ideas. Can you rewrite this to reflect his thinking more accurately?
  13. “University teachers at UJI have demonstrated to define course objectives through the lens of sustainability”. This does not make sense. Needs to be rewritten.
  14. The conclusion is not a proper conclusion. You are reporting data in the conclusion. That data should be in the findings section. A conclusion is a synthesis of your findings and what it contributes to the field. This needs a major rewrite. You might want to include limitations and also recommendations that come out of the research. A very poor conclusion that needs work.

Author Response

Please read the document attached.

Thank you,

Reviewer 4 Report

The manuscript has considerably been improved. My major concern is the connection between the TMTAS model and the previous theoretical framework which answers your first aim. I see them as two different frameworks and I do see why you need both in one manuscript. Or how are they connected to be presented in one manuscript? 

In the pdf, you find some comments, too. 

Author Response

Please, read the document attached.

Thank you
